# Volume Phase Transitions of Heliconical Cholesteric Gels under an External Field along the Helix Axis

**DOI:** 10.3390/gels6040040

**Published:** 2020-11-16

**Authors:** Akihiko Matsuyama

**Affiliations:** Department of Physics and Information Technology, Faculty of Computer Science and Systems Engineering, Kyushu Institute of Technology, Kawazu 680-4, Iizuka, Fukuoka 820-8502, Japan; matuyama@bio.kyutech.ac.jp

**Keywords:** liquid crystal, cholesteric gel, elastomer, volume phase transition

## Abstract

We present a mean field theory to describe cholesteric elastomers and gels under an external field, such as an electric or a magnetic field, along the helix axis of a cholesteric phase. We study the deformations and volume phase transitions of cholesteric gels as a function of the external field and temperature. Our theory predicts the phase transitions between isotropic (*I*), nematic (*N*), and heliconical cholesteric ($Ch_{H})$ phases and the deformations of the elastomers at these phase transition temperatures. We also find volume phase transitions at the $I-Ch_{H}$ and the $N-Ch_{H}$ phase transitions.

## 1. Introduction

Gels can undergo a volume phase transition with varying some parameters such as temperature, the degree of ionization, the pH, etc. [1,2]. Anisotropic deformations and volume changes of liquid crystalline gels have also unique properties due to the coupling between rubber elasticity and liquid crystalline ordering [3,4,5,6,7,8,9,10,11,12,13,14,15,16,17]. Urayama et al. have observed that the nematic ordering induces the volume changes of the gel [9,10]. The volume phase transitions of liquid crystalline gels have been theoretically studied for nematic [11,12,13,14,15], smectic A [16], and cholesteric gels [17].

Cholesteric (Ch) elastomers or polymer-stabilized Ch liquid crystals are important for applications in mirrorless lasers, in display devices, etc. [3]. External fields and temperature lead the magnetic-induced cholesteric-nematic transitions [18,19] and shear-induced uncoiling of the helix [20]. Kim and Finkelmann [21] have observed that a side-chain Ch elastomer shows an oblate chain conformation in the cholesteric phase. Our previous theory is qualitatively consistent with the experimental results [17]. Anisotropic deformation of a Ch gel due to an external field has also been reported [22]. A sufficiently high electric field imposed along the helical axis drives a finite elongation along the field axis.

Figure 1 shows a schematic representation of a side-chain Ch liquid crystalline polymer between two crosslinks on a Ch elastomer (or gel). In the Ch phase without the external field E=0, the director n is twisted along the pitch axis (*z* axis) with a pitch length *p* and the director is perpendicular to the pitch axis (Figure 1a). When the external field, such electric or magnetic fields, applies along the helix axis p of a Ch phase, the director n rotates along the pitch axis with a cone angle ϵ (Figure 1b). This phase is called a heliconical cholesteric ($Ch_{H}$) phase. Such an oblique helicoidal cholesteric phase induced by the electric field has been observed in banana-shaped liquid crystalline molecules [23].

In this paper, we theoretically study heliconical deformations of Ch elastomers and gels under the external field imposed along the helix axis of a Ch phase. We here focus on the deformations of Ch “elastomers” without solvent molecules and the swelling behaviors of Ch “gels” immersed in isotropic solvent molecules. We predict a rich variety of phase transitions of the Ch gels and elastomers induced by the external fields.

In the following, we show some numerical results of anisotropic deformations of $Ch_{H}$ elastomers (Section 2.1) and volume phase transitions of the gels (Section 2.2). In Section 3, we give a summary of this paper. Based on the neoclassical rubber theory of a nematic gel [3,17] and the free energy of a $Ch_{H}$ phase [24,25], we show the elastic free energy of a $Ch_{H}$ gel under the external field in Section 4.

## 2. Results and Discussions

In this section, we numerically calculate the equilibrium values of the swelling ratio $\alpha_{e}$ (or ϕ), the deformations $\kappa_{i}$ (i=x,y,z), and the order parameters. We here introduce the reduced-temperature $\tau =1/\nu_{L}$. For the numerical calculations, we set n=100, $n_{m}=n_{s}=2$, and $Q_{0}=0.06$. We also take the solvent molecule as a good solvent condition χ=0 in Equation (Equation 5).

In the following Section 2.1, we first show the deformations of the cholesteric elastomer melt without solvent molecules. In the next Section 2.2, we discuss the volume phase transitions of the heliconical cholesteric gel, immersed in solvent molecules.

### 2.1. Deformations of Cholesteric Elastomers under the External Field

In this subsection, we discuss deformations of a side-chain cholesteric elastomer melt without solvent molecules, where the elastomer has the constant volume $V=R_{0}^{3}N_{g}$ in Equation (Equation 1). Then the volume fraction of the melt elastomer is given by $\phi =1/\sqrt{n}$. The order parameters *S*, σ, and *Q* (or the deformation $\kappa_{z}$) are determined by the coupled-Equations (Equation 45), (Equation 49), and (Equation 51) as a function of the temperature τ.

Figure 2 shows the order parameters σ and *S* (a) and the deformations $\kappa_{i}$ (b), plotted against the external field $h_{L}$ at a temperature $T/T_{CI}=0.9$, where $T_{CI}$ is the temperature of the first-order $Ch_{H}-I$ phase transition. As increasing the external field $h_{L}$, the order parameter σ decreases and the cone angle ϵ of the $Ch_{H}$ phase decreases. At the intermediate stage of $h_{L}$, we find $\sigma \propto 1/h_{L}$. When σ=0, the $Ch_{H}$ elastomer changes to the N phase. When $h_{L}=0$, we have the cholesteric phase with σ=1. For a weak external field, the deformation $\kappa_{z}$ is smaller than $\kappa_{x}$ and the elastomer shows the oblate chain conformation, which is a spontaneous compression in the pitch axis (*z* direction). As increasing the external field, the value of $\kappa_{z}$ increases and the elastomer is elongated along the external field, where the elastomer has the prolate chain conformation with $\kappa_{z}>\kappa_{x}$. It has been experimentally observed that a sufficiently high electric field imposed along the helical axis drives a finite elongation exceeding 30% along the field axis [22].

Figure 3 shows the order parameters σ and *S* (a) and the deformations $\kappa_{z}$ (b) under the weak external field $h_{L}=0.01$ plotted against the temperature $T/T_{CI}$. At hight temperatures of $T/T_{CI}>1$, the value of *S* is small and it corresponds to the para-nematic phase (pN), or weak nematic phase. We here refer to it as the I(pN) phase because of S∼0. At $T/T_{CI}=1$, we have the first-order $Ch_{H}$
−I(pN) phase transition, where the order parameters *S* and σ jump. As shown in Figure 3b, the values of the deformations $\kappa_{i}$ jump at $T=T_{CI}$ and the value of $\kappa_{z}$ decreases with decreasing temperature. Near the phase transition temperature, the value of the order parameter σ is small and then the elastomer extends to parallel to the pitch axis (*z* axis): $\kappa_{z}>1$ and $\kappa_{x}=\kappa_{y}<1$. It corresponds to an prolate shape of the elastomer. However, on decreasing temperature, the value of σ increases and the elastomer causes the spontaneous compression ($\kappa_{z}<1$) in the pitch axis and elongates equally in the *x* and *y* directions ($\kappa_{x}=\kappa_{y}>1$). We then have $\kappa_{z}<\kappa_{x}$ and the elastomer shows an oblate shape at low temperatures. When σ=2/3 at $T/T_{CI}≃0.94$, we have $\kappa_{z}=\kappa_{y}=\kappa_{x}=1$. The shape of the $Ch_{H}$ elastomer is changed from prolate with σ<2/3 to oblate σ>2/3 with decreasing temperature. The pitch wavenumber *Q* of the $Ch_{H}$ phase is inversely proportional to $\kappa_{z}$: see Equation (Equation 39). As discussed in the previous paper [17], when $h_{L}=0$ without the external field, we have $\kappa_{z}<\kappa_{x}$ and the elastomer shows an oblate shape.

Figure 4 shows the order parameters σ and *S* (a) and the deformations $\kappa_{z}$ (b) under the stronger external field $h_{L}=0.15$ plotted against the temperature $T/T_{NI}$, where $T_{NI}$ is the temperature of the first-order phase transition. As the external field increases, the *N* phase with S>0 and σ=0 appears between the $Ch_{H}$ and I(pN) phases. We have the first-order N−I(pN) phase transition. We also have the second-order $Ch_{H}-N$ phase transition at $T/T_{NI}≃0.96$, where the order parameter *y* continuously increases. As shown in Figure 4b, in the *N* phase with σ=0, the elastomer is elongated along the pitch axis: $\kappa_{z}>1$, $\kappa_{x}=\kappa_{y}<1$, and the value of $\kappa_{z}$ increases with decreasing temperature, as discussed in Equation (Equation 50). In the $Ch_{H}$ phase, with decreasing temperature, the value of the order parameter σ increases and the value of $\kappa_{z}$ decreases. We then have a peak in the deformation curve of the elastomer at the $Ch_{H}-N$ phase transition. Further increasing the external field $h_{L}$, the $Ch_{H}-N$ phase transition temperature shifts to lower temperatures. We find the variety of the deformation curve of the cholesteric elastomers depending on the external fields.

### 2.2. Volume Phase Transitions of Cholesteric Gels under the External Field

In this subsection, we discuss the volume phase transitions and the deformations of the cholesteric gel under the external field. To derive the equilibrium swelling ϕ (or $\alpha_{e}$) of the gel, we numerically solve Equation (Equation 56).

Figure 5 shows the osmotic pressure Π (a) and the order parameters (b) plotted against the volume fraction ϕ of the gel for $h_{L}=0.05$ and $T/T_{NI}=1$, where the first-order N−I phase transition takes place in the gel. The osmotic pressure increases with increasing ϕ and jump at ϕ≃0,4, where the order parameter *S* jumps. Further increasing ϕ, the osmotic pressure curve has a minimum and jumps at ϕ≃0.86, where the order parameter σ jumps. The equilibrium value of ϕ is determined by the Maxwell construction. When $T/T_{NI}=1$, the area (1) and the area (2) become equal and then we have the first-order N−I phase transition takes place in the gel. The closed circles show the equilibrium volume fraction of the gel at $T/T_{NI}=1$, satisfying Π=0. The region (∂Π/∂ϕ)<0 corresponds to the unstable spinodal region [15]. Depending on the temperature, the curve of the osmotic pressure is changed. When $T/T_{NI}>1$, the area (1) becomes larger than that of the area (2) and then the *I* phase becomes stable. On the other hand, when $T/T_{NI}<1$, the area (2) becomes larger than that of the area (1) and then the *N* phase becomes stable or metastable. The equilibrium volume fraction of the gel can be obtained by solving Π=0 as a function of the temperature. In the following we show some numerical results of the swelling behaviors of the gel.

Figure 6 shows the volume fraction of the gel (a) and the order parameters (b) plotted against the temperature $T/T_{CI}$ under the weak external field $h_{L}=0.01$. The solid curves show the values of the equilibrium state and the dashed-curves correspond to the unstable region. At high temperatures of $T>T_{CI}$, the gel is in an isotropic state with S≃σ≃0 and is swollen with solvent molecules. With decreasing temperature, we find the discontinuous, or the first-order, volume phase transition from the swollen isotropic gel (ϕ≪1) to the condensed gel (ϕ∼1) at $T/T_{CI}=1$, where the order parameters jump and the $Ch_{H}$ phase with S>0 and σ>0 appears. Figure 7 shows the deformations $\kappa_{i}\left(i=x,y,z\right)$ of the gel against the temperature. In the *I* phase, the gel has a swollen isotropic state and we have $\kappa_{x}=\kappa_{y}=\kappa_{z}=1.36$. In the condensed-$Ch_{H}$ phase, we find $\kappa_{z}>\kappa_{x}$ because of σ<2/3, as discussed in Figure 3b.

Figure 8 shows the volume fraction of the gel (a) and the order parameters (b) plotted against the temperature $T/T_{NI}$ under the strong external field $h_{L}=0.05$. The solid curves show the values of the equilibrium state and the dashed-curves correspond to the unstable region. At high temperatures of $T>T_{NI}$, the gel is in an isotropic state with S≃σ≃0 and is swollen with solvent molecules. With decreasing temperature, we find the the first-order volume phase transition from the swollen isotropic gel (ϕ≪1) to the condensed nematic gel (ϕ∼1) at $T/T_{NI}=1$, where the orientational order parameter *S* jump and the *N* phase with S>0 and σ=0 appears. The osmotic pressures Π at $T/T_{NI}=1$ are shown in Figure 5a. We also have the first-order phase transition between *N* and $Ch_{H}$ phases at $T/T_{NI}≃0.962$, where the order parameters σ and *S* jump. The volume fraction of the $Ch_{H}$ gel slightly decrease, compared to the *N* gels. Figure 9 shows the deformations $\kappa_{i}\left(i=x,y,z\right)$ of the gel against the temperature. In the *I* phase, the gel has a swollen isotropic state and we have $\kappa_{x}=\kappa_{y}=\kappa_{z}=1.36$. In the *N* phase, the condensed gel is elongated parallel to the external field and shrink perpendicular to the field. We then have $\kappa_{z}>1$ and $\kappa_{z}<1$. In the $Ch_{H}$ phase, the gel causes spontaneous compression in the pitch axis. Note that the volume of the $Ch_{H}$ gel is almost the same as that of the *N* gel, while the shape of the gel is drastically changed through the $N-Ch_{H}$ phase transition. As discussed in the elastomers of Figure 4, the $N-Ch_{H}$ phase transition of the elastomer is the continuous phase transition, while the gel can be changed drastically in shape by using the solvent molecules.

## 3. Summary

We have presented a mean field theory to describe the deformations of cholesteric elastomers and gels under the external field imposed along the helix axis of a cholesteric phase. We calculate the deformation of cholesteric elastomers and volume phase transitions of cholesteric gels under the external field. Our theory demonstrates that a high external field imposed along the helical axis drives a finite elongation of the elastomer along the external field. The results can qualitatively describe the experimental results of a cholesteric gel. The deformation is given by Equation (Equation 50) as a function of σ(∝1/E) and *S* (Figure 2).

For a weak external field, we predict $I-Ch_{H}$ phase transition and the first-order volume phase transitions at the phase transition temperature $T_{CI}$. When $T>T_{CI}$, the gel is in a swollen isotropic state. Below $T<T_{CI}$, the gel is condensed with a heliconical cholesteric phase. For a strong external field, we predict the $I-N-Ch_{H}$ phase transitions and the volume phase transitions. At the $N-Ch_{H}$ phase transition temperature, the nematic gel is compressed along the helical axis due to the heliconical director in the $Ch_{H}$ phase. Our theory predicts a rich variety of volume phase transitions of $Ch_{H}$ gels.

## 4. Free Energy of Cholesteric Gels under an External Field

Consider a side-chain cholesteric liquid crystalline gel immersed in isotropic solvent molecules. The liquid crystalline “subchain” between two crosslinks in a network has the number *n* of segments. The repeating unit on the subchain consists of a rigid side-chain liquid crystalline molecule with the axial ratio $n_{m}$ and a flexible spacer with the number $n_{s}$ of segments, as shown in Figure 1c. Let *L* and *D* be the length and the diameter of the mesogen, respectively, and the axial ratio of the mesogen is given by $n_{m}=L/D$. The volume of the mesogen and that of the flexible spacer is given by $v_{m}=\left(\pi /4\right)D^{2}L$ and $v_{s}=a^{3}n_{s}$, respectively, where $a^{3}$ is the volume of a segment on the spacer. Then the volume of the subchain is given by $a^{3}n=a^{3}\left(n_{m}+n_{s}\right)t$, where *t* is the number of the repeating units on the subchain and we put $a^{3}=\left(\pi /4\right)D^{3}$. Let $N_{g}$ and $N_{0}$ be the number of the subchains and the solvent molecules inside the gel, respectively. The volume fraction of the gel is given by
(1)$$\phi =a^{3}nN_{g}/V,$$
where $N_{t}\left(=nN_{g}+N_{0}\right)$ is the total number of the segments inside the gel and $V=a^{3}N_{t}$ is the volume of the gel. We here assume that the volume per solvent molecule is the same volume $a^{3}$ as that of the segment on the subchain. The volume fraction of the mesogens is given by
(2)$$\phi_{m}=x_{m}\phi ,$$
where $x_{m}\equiv n_{m}/\left(n_{m}+n_{s}\right)$. Using the length $R_{i}$ of the subchain along the i(=x,y,z) axis, the volume occupied by the subchain is given by $R_{x}R_{y}R_{z}=V/N_{g}$. In this paper, we consider “uniaxial” deformations of the gel along the pitch axis p (parallel to *z* axis) and then we take $R_{x}=R_{y}$. The swelling of the gel can be characterized by
(3)$$\alpha_{e}=V/V_{0}=\phi_{0}/\phi ,$$
where $V_{0}$ is the initial volume of the gel and $\phi_{0}$ is the volume fraction of the gel in the initial state.

The free energy of the cholesteric gel under the external field is given by
(4)$$F=F_{mix}+F_{el}+F_{LC},$$
where the first term shows the free energy for an isotropic mixing of a gel and solvent molecules. According to the Flory–Huggins theory for polymer solutions, the free energy of the mixing is given by
(5)$$a^{3}\beta F_{mix}/V=\left(1-\phi \right)\ln\left(1-\phi \right)+\chi \phi \left(1-\phi \right),$$
where $\beta \equiv 1/k_{B}T$: *T* is the absolute temperature and $k_{B}$ Boltzmann constant, χ shows the isotropic (Flory–Huggins) interaction parameter between the gel and the solvent molecule. The second term in Equation (Equation 4) shows to the elastic free energy and the third term is the free energy of a heliconical cholesteric phase under an external field. In the following, we derive these free energies.

### 4.1. Elastic Free Energy of a Heliconical Cholesteric Phase

The elastic free energy $F_{el}$ comes from the deformation of the subchains on the gels. Based on the neoclassical rubber theory [3,17], it is given by
(6)$$\begin{array}{c} \beta F_{el}=\frac{1}{2}N_{g}\left(\lambda_{xx}^{2}+\lambda_{yy}^{2}+\lambda_{zz}^{2}-3-\ln\frac{a^{3}}{{ł}_{xx}{ł}_{yy}{ł}_{zz}}\right), \end{array}$$
where $\lambda_{ii}$ is the strain of the gel. The strain tensor is given by [6,11,13]
(7)$$\begin{array}{c} \lambda_{ii}=R_{i}/R_{i0}, \end{array}$$
where $R_{i0}$ is the spontaneous mean-square radius of the subchain along the i(=x,y,z) axis. Using the effective step (bond) length tensor ${ł}_{ii}$ of an anisotropic Gaussian chain, we have
(8)$$\begin{array}{c} R_{i0}^{2}={ł}_{ii}an=R_{0}^{2}\left(\frac{{ł}_{ii}}{a}\right), \end{array}$$
where $R_{0}\equiv a\sqrt{n}$ is the spontaneous radius of an ideal chain. When ${ł}_{ii}=a$, we have an isotropic Gaussian chain. The effective step length tensor ${ł}_{ij}$ around the director field n is given by [3]
(9)$${ł}_{ij}={ł}_{⊥}\delta_{ij}+\left({ł}_{∥}-{ł}_{⊥}\right)n_{i}n_{j},$$
where $n_{i}$ is the element *i* of the director and ${ł}_{∥}\left({ł}_{⊥}\right)$ is the step length parallel (perpendicular) to the director. In the uniaxial nematic elastomers, the average shape of the subchains (backbone) is anisotropic and elongated along the nematic director n. The side-chain elastomers have different conformations, depending on the type of linking the mesogens to the backbone or spacer. We here assume the side-on linking as sown in Figure 1c, similar to the main-chain case. There are many complexities in the difference between the main- and side-chain elastomers, however, we assume that the main contribution to the elastic energy is the deformation of the subchain: $R_{i0}$.

In this paper, we consider a longitudinal external field parallel to the pitch axis p (the *z* axis) as shown in Figure 10. Then, the electric field E is given by
(10)E=(0,0,E),
where *E* shows the strength of the external field. When the dielectric anisotropy is positive: $\Delta \epsilon_{a}>0$, the liquid crystal molecules tend to orient along the external field. The longitudinal fields, or longitudinal deformations, along the helical axis *z* can swing the director along the pitch axis p. With the cone angle ϵ of the director n measured from the pitch axis p, the director rotates out of the perpendicular plane onto the surface of the cone angle ϵ as shown in Figure 10. In this conical state, the director is given by
(11)n(z)=(sinϵcosω(z),sinϵsinω(z),cosϵ),
where the director is uniformly twisted along the *z* axis with the pitch p=2π/|q| and the azimuthal angle ω is given as a function of the position *z*: ω=qz. When the pitch wavenumber q>0
(q<0), we have a right (left)-handed helix. The cone angle ϵ is a constant and does not depend on the position *z*, as shown in Figure 1b.

Substituting Equation (Equation 11) into (Equation 9), we obtain
(12)$${ł}_{xx}={ł}_{⊥}+\left({ł}_{∥}-{ł}_{⊥}\right){\langle \sin^{2}\epsilon \rangle }_{l}{\langle \cos^{2}qz\rangle }_{l},$$
(13)$${ł}_{yy}={ł}_{⊥}+\left({ł}_{∥}-{ł}_{⊥}\right){\langle \sin^{2}\epsilon \rangle }_{l}{\langle \sin^{2}qz\rangle }_{l},$$
(14)$${ł}_{zz}={ł}_{⊥}+\left({ł}_{∥}-{ł}_{⊥}\right){\langle \cos^{2}\epsilon \rangle }_{l}.$$
The director n is given by the average orientation of mesogens. The spacer chains are flexible and then the step length $l_{ij}$ is given by the average orientation of mesogens. The averages ${\langle \cdots \rangle }_{l}$ of Equations (Equation 12)–(Equation 14) can be given by the average of the local director. The length ${ł}_{∥}\left({ł}_{⊥}\right)$ is the step length parallel (perpendicular) to the uniaxial deformation. For the uniaxial deformation of the gel, the average over the azimuthal angle ω on the x−y plane is given by
(15)$${\langle \cos^{2}qz\rangle }_{l}={\langle \sin^{2}qz\rangle }_{l}=1/2.$$

We here consider a random walk (the freely-jointed model) [3,11,13] with the bond length acosθ along the director and the length asinθ in the perpendicular direction to the director, where θ is the angle between the director n and the orientation Ω of the mesogen as shown in Figure 10. We then have [3]
(16)$${ł}_{∥}=3a\langle \cos^{2}\theta \rangle =a\left(1+2S\right),$$
(17)$${ł}_{⊥}=\frac{3}{2}a\langle \sin^{2}\theta \rangle =a\left(1-S\right),$$
where $S=\left(3/2\right)(\langle \cos^{2}\theta \rangle -1/3)$ is the scalar orientational order parameter of mesogens. Equations (Equation 16) and (Equation 17) are given as a function of the average orientational order parameter S for all mesogens. In the isotropic phase, the average is given by $\langle \cos^{2}\theta \rangle =1/3$ and $\langle \sin^{2}\theta \rangle =2/3$ and then we have ${ł}_{∥}={ł}_{⊥}=a$. Substituting Equations (Equation 15)–(Equation 17) into (Equation 12)–(Equation 14), we obtain
(18)$${ł}_{xx}={ł}_{yy}=a\left(1-S+\frac{3}{2}S\sigma \right),$$
(19)$${ł}_{zz}=a\left(1+2S-3S\sigma \right),$$
where $\sigma =\langle \sin^{2}\epsilon \rangle$ is the order parameter of the heliconical cholesteric phase. When σ=0, we have a nematic phase and when σ=1 we have a cholesteric phase.

The volume fraction ϕ of the gel is given by
(20)$$\phi =\frac{a^{3}n}{R_{z}R_{x}^{2}}=\frac{1}{\sqrt{n}\kappa_{z}\kappa_{x}^{2}},$$
where we define the deformation ratio ($\kappa_{i}$) related to an isotropic Gaussian chain:(21)$$\kappa_{z}\equiv R_{z}/R_{0},$$
(22)$$\kappa_{x}\equiv R_{x}/R_{0}=R_{y}/R_{0},$$
and we then have
(23)$$\kappa_{x}^{2}=\frac{1}{\sqrt{n}\phi \kappa_{z}}.$$
Using $\kappa_{z}$, the strain $\lambda_{ii}$ (Equation (Equation 7)) is given as a function of the order parameters:(24)$$\lambda_{xx}=\lambda_{yy}={\left(1-S+\frac{3}{2}S\sigma \right)}^{-1/2}{\left(\sqrt{n}\phi \kappa_{z}\right)}^{-1/2}.$$
(25)$$\lambda_{zz}={\left(1+2S-3S\sigma \right)}^{-1/2}\kappa_{z}.$$

Substituting Equations (Equation 24) and (Equation 25) into (Equation 6), we finally obtain the elastic free energy of heliconical cholesteric elastomers:(26)$$\begin{array}{ccc} a^{3}\beta F_{el}/V & = & \frac{\phi }{2n}[\frac{\kappa_{z}^{2}}{1+2S-3S\sigma }+\frac{2}{\sqrt{n}\phi \left(1-S+\frac{3}{2}S\sigma \right)\kappa_{z}}\\ & & -3+\ln A], \end{array}$$
where we define
(27)$$\begin{array}{ccc} A & = & l_{xx}l_{yy}l_{zz}/a^{3}\\ & = & {\left(1-S+\frac{3}{2}S\sigma \right)}^{2}\left(1+2S-3S\sigma \right). \end{array}$$
When σ=0, Equation (Equation 26) results in the elastic free energy of the cholesteric gels [17].

The configuration of the mesogens on the subchains is characterized by its position vector r and its orientation unit vector Ω, defined by the solid angle dΩ(=sinθdθdφ), as shown in Figure 10. Let f(n(r)·Ω) be the orientational distribution function of the mesogens, where n(r) is the local director. The orientational order parameter of the mesogens is given by
(28)$$\begin{array}{c} S=\int P_{2}\left(\cos\theta \right)f\left(n\left(r\right)\cdot \Omega \right)d\Omega , \end{array}$$
where $P_{2}\left(\xi \right)=\left(3/2\right)\left(\xi^{2}-1/3\right)$ is the second Legendre polynomials with ξ=cosθ.

### 4.2. Free Energy of a Heliconical Cholesteric Phase under an External Field

In this subsection, we introduce the free energy $F_{LC}$ of heliconical cholesteric phase under an external field. For these free energy, we can use the free energy of the Ch phase under the external field, which has been discussed in our previous paper [17]. The free energy $F_{LC}$ in Equation (Equation 4) consists of three terms:(29)$$F_{LC}=F_{nem}+F_{d}+F_{ext}.$$
The first term is the usual nematic free energy of Maier–Saupe type:(30)$$\begin{array}{ccc} a^{3}\beta F_{nem}/V & = & \frac{\phi_{m}}{n_{m}}\int f\left(n\left(r\right)\cdot \Omega \right)\ln4\pi f\left(n\left(r\right)\cdot \Omega \right)d\Omega \\ & & -\frac{1}{2}\nu_{L}\phi_{m}^{2}S^{2}, \end{array}$$
where the parameter $\nu_{L}=-\beta U_{2}\left(>0\right)$ shows a nematic interaction, which has been used in Maier–Saupe theory [26]. We here assume that the interaction potentials $U_{2}$ between mesogens is the short range $d_{0}$ of the order of the diameter of the mesogen.

The mesogens are bounded to the polymer backbone. However, the backbone chains are flexible and have many conformations. As a result, the mesogens bounded to the backbone chains can move with the backbone chain and can behave like to freely rotate. The rotation of mesogens are restricted due to being bounded to the backbone chain, however, the decrease in the entropy of rotation of the mesogens is less and the mesogens have the large contribution of the rotational entropy because the mesogens can move with the backbone chain. We here assume that the interaction parameter $\nu_{L}$ includes the effects of these constrained mesogens. Then the orientational (nematic) bulk free energy is given by the orientations of the mesogens. However, the “translational degrees of freedom” of the mesogens (or rods) are restricted due to bounded to the backbone chain. Then the translational entropy of mesogens are not included in the mixing free energy ($F_{mix}$). The center of gravity of the gel is fixed. The swelling or shrink of the gel is promoted by the translational degrees of freedom of the solvent molecules.

The second term in Equation (Equation 29) is the distortion free energy of the $Ch_{H}$ phase due to the spatial variations of the director. We here introduce the tensor order parameter [18]
(31)$$Q_{\alpha \beta }\left(r\right)=S\left(\frac{3}{2}n_{\alpha }\left(r\right)n_{\beta }\left(r\right)-\frac{1}{2}\delta_{\alpha \beta }\right),$$
where $n_{\alpha }$ is the α(=x,y,z) component of the director n and $\delta_{\alpha \beta }$ is the Kronecker delta function. Taking into account the chiral interactions between mesogens, [27] the distortion free energy, including the first and second spatial derivatives of the tensor order parameter, is given by [17,28]
(32)$$\begin{array}{ccc} a^{3}\beta F_{d}/V & = & \frac{1}{2}\nu_{L}\phi_{m}^{2}\frac{1}{9}\partial_{\gamma }Q_{\alpha \beta }\left(r\right)\partial_{\gamma }Q_{\alpha \beta }\left(r\right)d_{0}^{2}\\ & - & \frac{1}{2}c_{L}\phi_{m}^{2}\frac{4}{9}\epsilon_{\alpha \beta \gamma }Q_{\mu \beta }\left(r\right)\partial_{\alpha }Q_{\mu \gamma }\left(r\right)d_{0}, \end{array}$$
where $\epsilon_{\alpha \beta \gamma }$ is a Levi–Civita antisymmetric tensor of the third rank and $\partial_{\kappa }=\partial /\partial r_{\kappa }$ is the first spatial derivative of the tensor order parameter. The parameter $c_{L}=-\beta U_{1}$ shows a chiral pseudoscalar interaction between the liquid crystal molecules. The positive (negative) value of the $c_{L}$ means a left (right)-handed helix.

The last term in Equation (Equation 29) is the free energy of electric (or magnetic) external fields relevant to an orientational order. We here consider the coupling between the nematic director and the external field. When the external electric field E is applied to the liquid crystal molecules, having a dielectric anisotropy $\Delta \epsilon_{a}$, the external free energy is given by [18]
(33)$$\begin{array}{ccc} F_{ext} & = & -\phi_{m}\epsilon_{0}\Delta \epsilon_{a}\int E_{\alpha }Q_{\alpha \beta }\left(r\right)E_{\beta }dr, \end{array}$$
where $E_{\alpha }$ is the α component of the external field E. An external magnetic field can also be treated the same way as the electric field.

Substituting Equations (Equation 10) and (Equation 11) into Equations (Equation 30), (Equation 32), and (Equation 33), we obtain the free energy of liquid crystalline phases including *N*, Ch, and $Ch_{H}$ phases:(34)$$F_{LC}=F_{nem}+F_{d}+F_{ext}=F_{nem}+F_{dis},$$
where we have separated the free energy into two terms for convenience [17]. One is the nematic free energy $F_{nem}$ of Maier–Saupe type [26] and the other is the distortion free energy ($F_{dis}=F_{d}+F_{ext}$) due to the spatial variation of the director under the external field. The dimensionless nematic free energy ($f_{nem}$) is given by
(35)$$\begin{array}{ccc} f_{nem} & = & a^{3}\beta F_{nem}/V\\ & = & \frac{\phi_{m}}{n_{m}}\int f\left(n\left(r\right)\cdot \Omega \right)\ln4\pi f\left(n\left(r\right)\cdot \Omega \right)d\Omega \\ & & -\frac{1}{2}\nu_{L}\phi_{m}^{2}S^{2}-\phi_{m}Sh_{L}^{2}, \end{array}$$
where we define $h_{L}^{2}=a^{3}\beta \epsilon_{0}\Delta \epsilon_{a}E^{2}$ for $\Delta \epsilon_{a}>0$ and the last term comes from the external free energy $F_{ext}$. Substituting the tensor order parameter Equation (Equation 31) into Equation (Equation 32), the dimensionless distortion free energy ($f_{dis}$) including Ch and ${\mathrm{Ch}}_{H}$ phases in Equation (Equation 34) is given by (see Appendix A)
(36)$$\begin{array}{ccc} f_{dis} & = & a^{3}\beta F_{dis}/V\\ & = & \frac{1}{2}\nu_{L}\phi_{m}^{2}S^{2}g\left(\sigma ,Q\right), \end{array}$$
where we define the distortion function
(37)$$\begin{array}{ccc} g(\sigma ,Q) & = & \frac{1}{2}\left({\tilde{k}}_{2}-{\tilde{k}}_{3}\right)Q^{2}\sigma^{2}\\ & & +\left[\left(\frac{1}{2}{\tilde{k}}_{3}Q-{\tilde{k}}_{2}Q_{0}\right)Q+\eta \right]\sigma , \end{array}$$
(38)$$\begin{array}{c} \eta =\frac{3h_{L}^{2}}{\nu_{L}\phi_{m}S}\propto E^{2}, \end{array}$$
and the pitch wavenumber $Q=qd_{0}$ [17]. The values of ${\tilde{k}}_{2}$ and ${\tilde{k}}_{3}$ correspond to the dimensionless twist and bend elastic constants of a pure mesogen, respectively. The value $Q_{0}\left(\equiv c_{L}/\nu_{L}\right)$ shows the pitch wavenumber of the pure Ch phase in the absence of the external field [28]. The distortion function g(σ,Q) can describe the N, Ch, and ${\mathrm{Ch}}_{H}$ phases of pure liquid crystalline molecules. The function g(σ,Q) has a minimum as a function of *Q* and σ for ${\tilde{k}}_{2}>{\tilde{k}}_{3}$. When σ=1, or ϵ=π/2, the bend term ${\tilde{k}}_{3}$ disappears and the usual Ch phase appears. When σ=0, or ϵ=0, we have the usual N phase because of g=0. Depending on the strength $h_{L}$ of the external field, we have N, Ch, and ${\mathrm{Ch}}_{H}$ phases. Note that the total free energy (Equation (Equation 4)) of our system is given by the sum of Equations (Equation 5), (Equation 35), and (Equation 36).

The pitch wavenumber *Q* depends on the deformation $\kappa_{z}$ along the pitch axis of the cholesteric gel. We here assume the “affine” deformation $p=p_{0}\kappa_{z}$ of the subchain and then the pitch length $p(\propto Q^{-1}$) is given as a function of the deformation ratio $\kappa_{z}$ [17]:(39)$$\begin{array}{c} Q=\frac{2\pi d_{0}}{p}=Q_{0}/\kappa_{z}. \end{array}$$

### 4.3. Orientational Distribution Function

In this subsection, we derive the equilibrium distribution function. The orientational distribution function f(n(r)·Ω) of the mesogens is determined by the free energy (Equation 4) with respect to this function: ${\left(\delta F/\delta f\left(\xi \right)\right)}_{\phi ,Q,\sigma }=0$, where ξ=cosθ, under the normalization condition
(40)$$\begin{array}{c} \int f(n(r)\cdot \Omega )d\Omega =1. \end{array}$$
We then obtain the distribution function of the mesogens (see Appendix B):(41)$$\begin{array}{c} f\left(\xi \right)=\frac{1}{Z}\exp\left[\Gamma P_{2}\left(\xi \right)\right], \end{array}$$
where we define
(42)$$\begin{array}{c} \Gamma =n_{m}\left[\nu_{L}\phi_{m}S\left(1-g(\sigma ,Q)\right)+h_{L}^{2}-B\left(S,\phi \right)\right], \end{array}$$
and
(43)$$\begin{array}{ccc} B(S,\phi ) & = & \frac{\left(1-\frac{3}{2}\sigma \right)}{nx_{m}}[\frac{\kappa_{z}^{2}}{{\left(1+2S-3S\sigma \right)}^{2}}\\ & & -\frac{1}{\sqrt{n}\phi {\left(1-S+\frac{3}{2}S\sigma \right)}^{2}\kappa_{z}}\\ & & -\frac{3(1-\frac{3}{2}\sigma )S}{\left(1+2S-3S\sigma \right)\left(1-S+\frac{3}{2}S\sigma \right)}]. \end{array}$$
The constant *Z* is determined by the normalization condition as $Z=4\pi I_{0}\left[S\right]$ and the function $I_{m}\left[S\right]$ is defined as
(44)$$\begin{array}{c} I_{m}\left[S\right]=\int_{0}^{1}{\left[P_{2}\left(\xi \right)\right]}^{m}\exp\left[\Gamma P_{2}\left(\xi \right)\right]d\xi , \end{array}$$
where m=0,1,⋯. Substituting Equation (Equation 41) into Equation (Equation 28), the orientational order parameter can be determined by
(45)$$\begin{array}{c} S=I_{1}\left[S\right]/I_{0}\left[S\right]. \end{array}$$
Using the distribution function (Equation (Equation 41)), the heliconical cholesteric free energy (Equation (Equation 34)) is given by
(46)$$\begin{array}{ccc} a^{3}\beta F_{LC}/V & = & \frac{1}{2}\nu_{L}\phi_{m}^{2}S^{2}\left(1-g(\sigma ,Q)\right)-\frac{\phi_{m}}{n_{m}}\ln I_{0}\left[S\right]\\ & & -\phi_{m}SB\left(S,\phi \right). \end{array}$$
The total free energy *F* is given by the sum of Equations (Equation 5), (Equation 26) and (Equation 46). Apparently, when S=0, or an isotropic phase, the free energy (Equation 46) becomes zero.

### 4.4. Determination of the Order Parameters σ and *Q*

The deformation of the cholesteric gel, or elastomer, at a thermal equilibrium state is determined by
(47)$${\left(\partial f_{e}/\partial Q\right)}_{\sigma ,S,\phi }=0,$$
and
(48)$${\left(\partial f_{e}/\partial \sigma \right)}_{Q,S,\phi }=0,$$
where we define the dimensionless free energy $f_{e}=a^{3}\beta F/V$ for convenience. Note that the deformation $\kappa_{z}$ is given as a function of *Q* through Equation (Equation 39) and then Equation (Equation 47) is the same as ${\left(\partial f_{e}/\partial \kappa_{z}\right)}_{\sigma ,S,\phi }=0$. From Equation (Equation 47), we obtain
(49)$$\begin{array}{ccc} & & \frac{\phi }{n}\left[\frac{\kappa_{z}}{1+2S-3S\sigma }-\frac{1}{\sqrt{n}\phi \left(1-S+\frac{3}{2}S\sigma \right)\kappa_{z}^{2}}\right]\\ & & +\frac{1}{2}\nu_{L}\phi_{m}^{2}S^{2}\sigma \left({\tilde{k}}_{2}-\left(\left({\tilde{k}}_{2}-{\tilde{k}}_{3}\right)\sigma +{\tilde{k}}_{3}\right)\frac{1}{\kappa_{z}}\right){\left(\frac{Q_{0}}{\kappa_{z}}\right)}^{2}=0, \end{array}$$
where the first two terms show the contribution from the elastic free energy and the last term comes from the distortion free energy. The coefficient of the last term shows the dimensionless twist elastic constant [28]: $\tilde{K_{22}}=\left(1/2\right)\nu_{L}\phi_{m}^{2}S^{2}$. For small $Q_{0}$, we can neglect the last term of Equation (Equation 49) and the deformation $\kappa_{z}$ is approximately given by
(50)$$\begin{array}{c} \kappa_{z}≃{\left(\frac{1+2\left(1-\frac{3}{2}\sigma \right)S}{\sqrt{n}\phi (1-\left(1-\frac{3}{2}\sigma \right)S}\right)}^{1/3}. \end{array}$$
In the *N* phase with σ=0 and S>0, the deformation ratio $\kappa_{z}$ increases with increasing the orientational order parameter *S*. While in the $Ch_{H}$ phase with σ>0 and S>0, the value of $\kappa_{z}$ decreases with increasing the order parameter σ and the elastomer (or gel) tends to compress parallel to the pitch axis. On the other hand, as increasing the external field *E*, the value of σ decreases and then the elastomer tends to elongate along the pitch axis. When σ=2/3, we find $\kappa_{z}≃{\left(1/\sqrt{n}\phi \right)}^{1/3}$, which corresponds to the value of the *I* phase without the external field. When σ=1, Equation (Equation 50) results in the deformation of the cholesteric elastomers, where the elastomer is compressed along the pitch axis [17]. From Equation (Equation 48), we obtain
(51)$$\begin{array}{ccc} & & \frac{3\phi }{2n}[\frac{S\kappa_{z}^{2}}{{\left(1+2S-3S\sigma \right)}^{2}}-\frac{S}{\sqrt{n}\phi {\left(1-S+\frac{3}{2}S\sigma \right)}^{2}\kappa_{z}}\\ & & +\frac{3S^{2}\left(1-\frac{3}{2}\sigma \right)}{\left(1+2S-3S\sigma \right)\left(1-S+\frac{3}{2}S\sigma \right)}]\\ & & +\frac{1}{2}\nu_{L}\phi_{m}^{2}S^{2}\left(({\tilde{k}}_{2}-{\tilde{k}}_{3}\right)Q^{2}\sigma +\left(\frac{1}{2}{\tilde{k}}_{3}Q-{\tilde{k}}_{2}Q_{0}\right)Q+\eta )=0, \end{array}$$
The order parameters σ and *Q* are numerically determined from Equations (Equation 49) and (Equation 51).

### 4.5. Equilibrium State of a Gel

The chemical potential of the solvent molecule is given by
(52)$$\begin{array}{ccc} a^{3}\beta \left(\mu_{0}-\mu_{0}^{\circ }\right) & = & a^{3}\beta {\left(\partial F/\partial N_{0}\right)}_{N_{g}}\\ & = & f_{e}-\phi \left(\partial f_{e}/\partial \phi \right), \end{array}$$
where $\mu_{0}$ shows the chemical potential of the solvent molecule inside the gel and $\mu_{0}^{\circ }$ is that of the pure solvent molecule outside the gel. The total free energy is given by $f_{e}\left(\phi ,Q,S,\sigma \right)$ where order parameters *Q*, σ, and *S* are given as a function of ϕ. Then the total derivative is
(53)$$\begin{array}{ccc} \left(\frac{df_{e}}{d\phi }\right) & = & {\left(\frac{\partial f_{e}}{\partial \phi }\right)}_{Q,\sigma ,S}+{\left(\frac{\partial f_{e}}{\partial Q}\right)}_{\phi ,\sigma ,S}\left(\frac{\partial Q}{\partial \phi }\right)\\ & & +{\left(\frac{\partial f_{e}}{\partial \sigma }\right)}_{\phi ,Q,S}\left(\frac{\partial \sigma }{\partial \phi }\right)+{\left(\frac{\partial f_{e}}{\partial S}\right)}_{\phi ,Q,\sigma }\left(\frac{\partial S}{\partial \phi }\right). \end{array}$$
We here evaluate *Q* (or $\kappa_{z}$), σ, and S determined above that ${\left(\partial f_{e}/\partial Q\right)}_{\phi ,S,\sigma }=0$ (Equation (Equation 47)), ${\left(\partial f_{e}/\partial \sigma \right)}_{\phi ,Q,S}=0$ (Equation (Equation 48)), and
(54)$$\begin{array}{c} {\left(\frac{\partial f_{e}}{\partial S}\right)}_{\phi ,Q,\sigma }={\left(\frac{\delta f_{e}}{\delta f(\xi )}\right)}_{\phi ,Q,\sigma }{\left(\frac{\partial S}{\partial f(\xi )}\right)}^{-1}=0, \end{array}$$
respectively. Equation (Equation 54) gives the equilibrium distribution function (Equation (Equation 41)). Thus, the total derivative in Equation (Equation 52) becomes ${\left(\partial f_{e}/\partial \phi \right)}_{Q,S,\sigma }$ and yields
(55)$$\begin{array}{ccc} a^{3}\beta \left(\mu_{0}-\mu_{0}^{\circ }\right) & = & \frac{1}{n\sqrt{n}\left(1-S+3S\sigma \right)\kappa_{z}}\\ & & +\ln\left(1-\phi \right)+\phi +\chi x_{m}^{2}\phi^{2}\\ & & +\frac{1}{2}\nu_{L}x_{m}^{2}\phi^{2}S^{2}\left(1-g(\sigma ,Q)\right). \end{array}$$
The equilibrium swelling $\alpha_{e}$ (or ϕ) of the gel can be determined by the balance of the chemical potentials (osmotic pressure) among the solvent molecules existing outside and inside the gel:(56)$$\begin{array}{c} \Pi =-a^{3}\beta \left(\frac{\partial F}{\partial V}\right)=-a^{3}\beta \left(\mu_{0}-\mu_{0}^{\circ }\right)=0. \end{array}$$
When the dimensionless osmotic pressure (Π) versus ϕ has the van der Waals loops, the equilibrium value of ϕ is determined by the Maxwell construction. The region
(57)$$\begin{array}{c} {\left(\frac{\partial \Pi }{\partial \phi }\right)}_{Q,\sigma ,S}=\phi {\left(\frac{\partial^{2}f_{e}}{\partial \phi^{2}}\right)}_{Q,\sigma ,S}<0 \end{array}$$
corresponds to an unstable spinodal region and ${\left(\partial \Pi /\partial \phi \right)}_{Q,\sigma ,S}>0$ corresponds to a stable (or metastable) region [12,15]. Then the equilibrium state of the gel follows the volume curve on the ϕ− Temperature plane, determined by the condition Π=0 with the Maxwell construction, which is equivalent to minimizing the free energy $f_{e}$ with respect to ϕ. Thus, in analogy with the gas–liquid phase transitions, we can discuss the isotropic-liquid crystal phase transitions of the gels, by evaluating the Π−ϕ curves.

## Figures and Tables

**Figure 1 gels-06-00040-f001:**
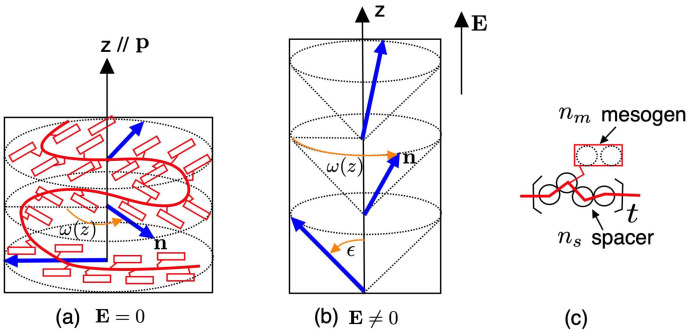
Schematic representation of a local part of a side-chain liquid crystalline polymer, or a subchain on a cholesteric elastomer (or gel). (**a**) A cholesteric phase without the external field E=0. The director n varies along the helical axis p, which is parallel to the *z* axis. (**b**) A heliconical cholesteric phase under an external field E≠0 applied along the pitch axis. The director n rotates along the pitch axis with a cone angle ϵ. (**c**) Model of the repeating unit on the subchain with the axial ratio $n_{m}\left(=2\right)$ and the spacer $n_{s}\left(=4\right)$ (see text for details).

**Figure 2 gels-06-00040-f002:**
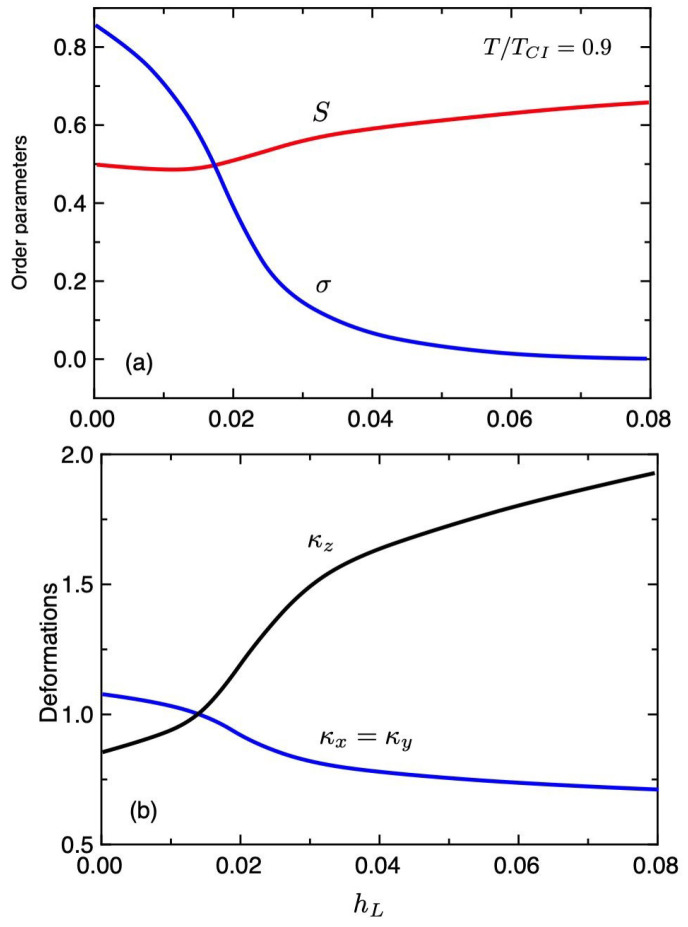
Order parameters *S* and σ (**a**) and the deformations $\kappa_{i}$ (**b**), plotted against the external field $h_{L}$ at a temperature $T/T_{CI}=0.9$.

**Figure 3 gels-06-00040-f003:**
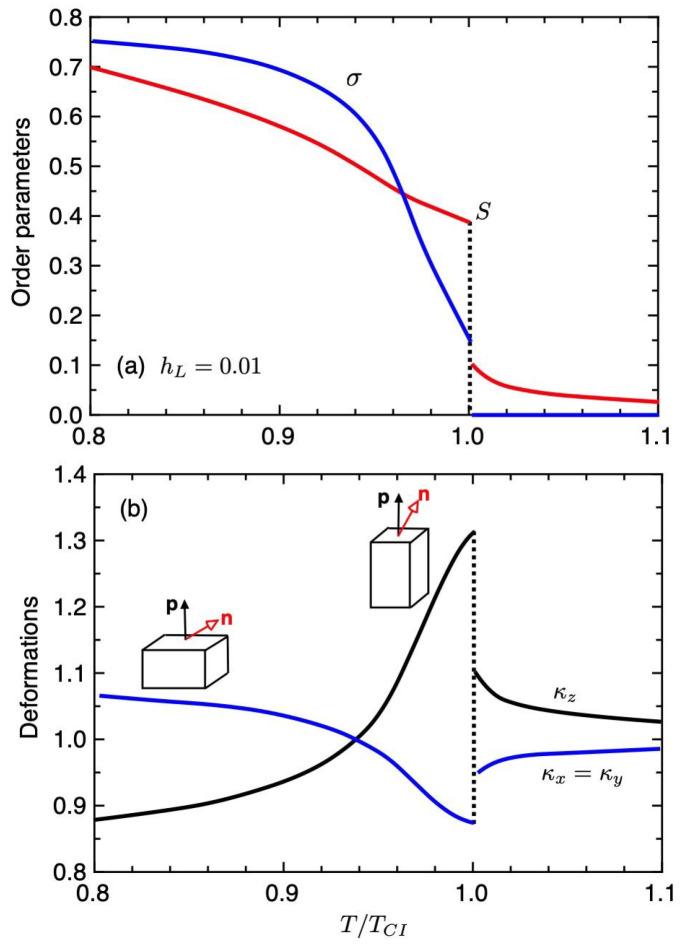
Order parameter *S* and σ as a function of the temperature (**a**) and deformations $\kappa_{i}$ under the external field $h_{L}=0.01$ plotted against the temperature $T/T_{CI}$, where $T_{CI}$ shows the temperature of the first-order $Ch_{H}-I\left(pN\right)$ phase transition (**b**). The cholesteric elastomer causes the spontaneous elongation in the pitch axis near the transition temperature $T_{CI}$. With decreasing temperature, the value of $\kappa_{z}$ decreases and the value of $\kappa_{x}$ increases.

**Figure 4 gels-06-00040-f004:**
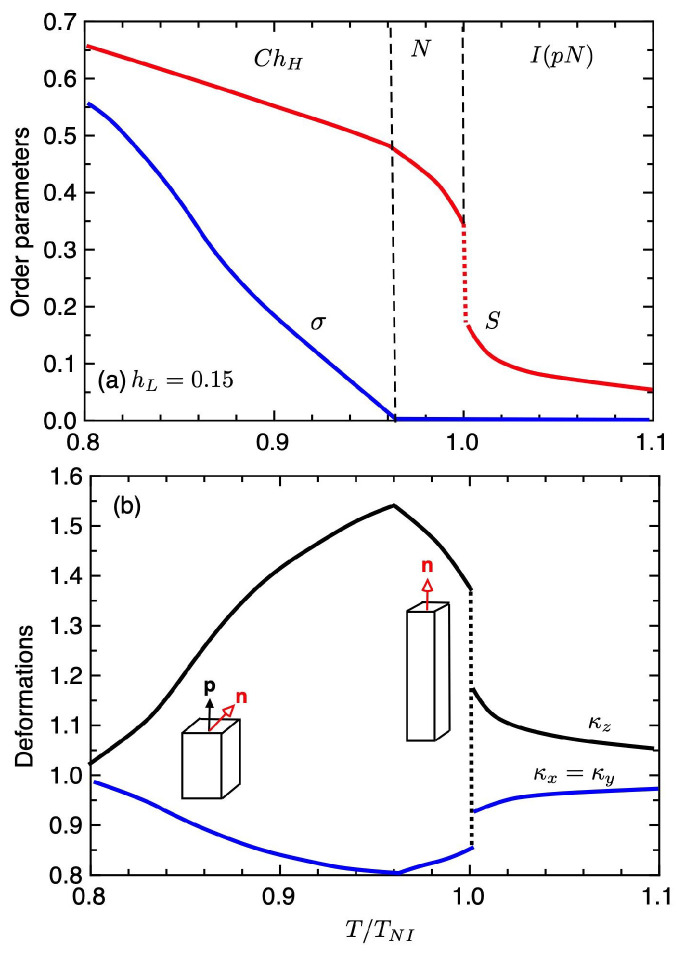
Order parameter *S* and σ as a function of the temperature (**a**) and deformations $\kappa_{i}$ under the strong external field $h_{L}=0.15$ plotted against the temperature $T/T_{NI}$, where $T_{NI}$ shows the temperature of the first-order N−I(pN) phase transition (**b**). The *N* phase with S>0 and σ=0 appears between the $Ch_{H}$ and I(pN) phases.

**Figure 5 gels-06-00040-f005:**
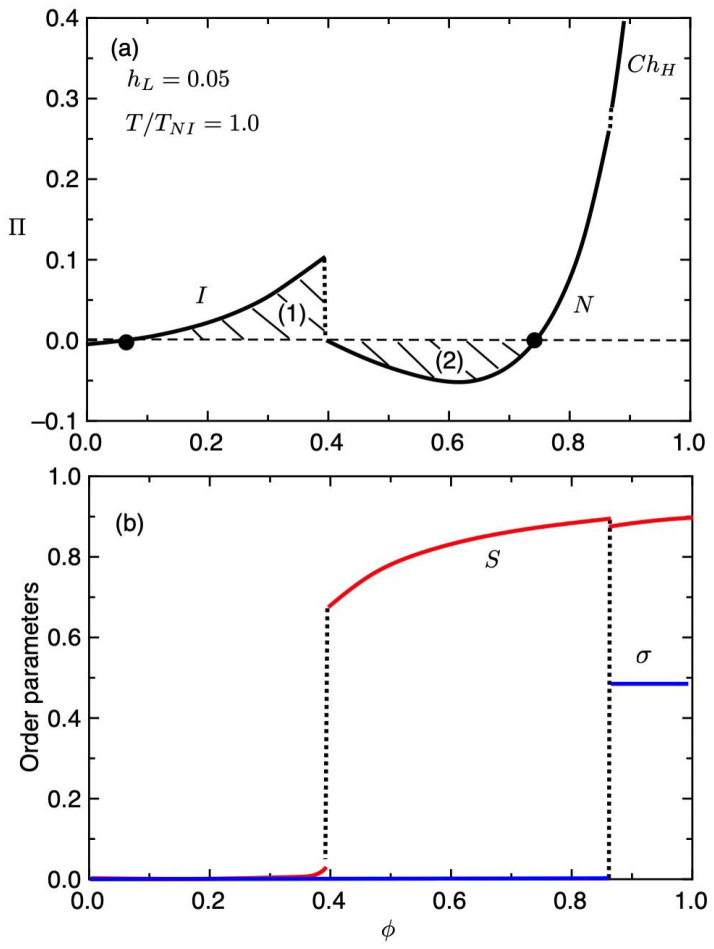
Dimensionless osmotic pressure Π (**a**) and the order parameters (**b**) plotted against the volume fraction ϕ of the gel for $h_{L}=0.05$ and $T/T_{NI}=1$, where the first-order N−I phase transition takes place in the gel.

**Figure 6 gels-06-00040-f006:**
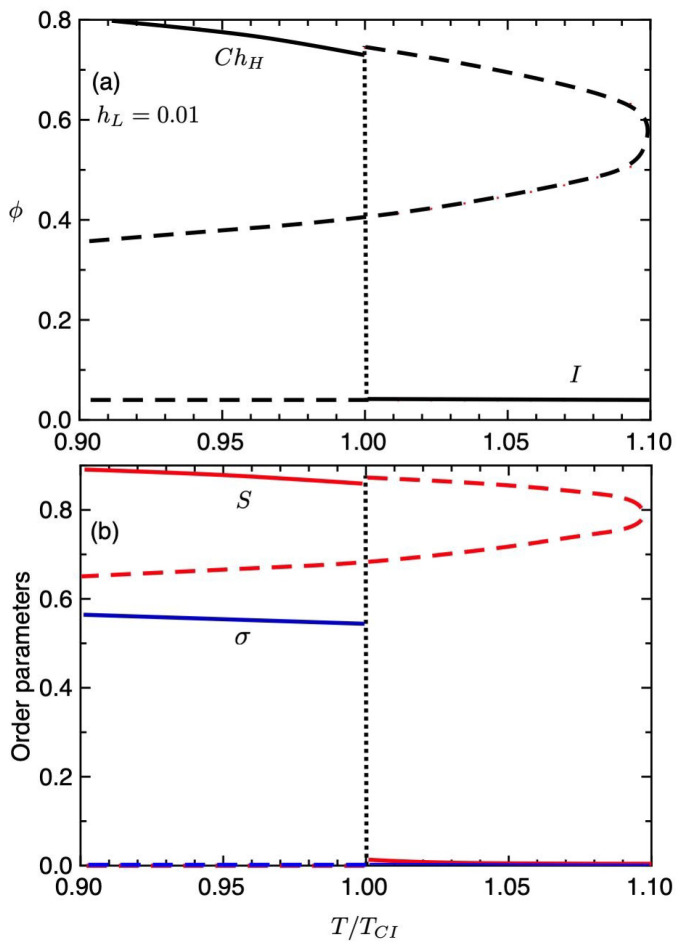
Equilibrium volume fraction of the gel (**a**) and the order parameters (**b**) plotted against the temperature $T/T_{CI}$ under the weak external field $h_{L}=0.01$.

**Figure 7 gels-06-00040-f007:**
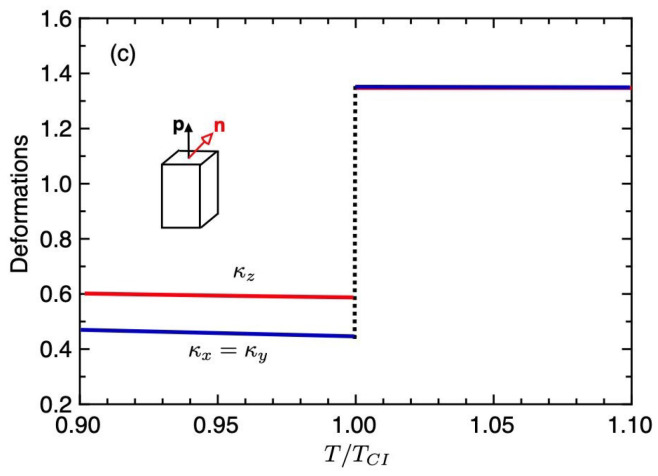
Deformations $\kappa_{i}\left(i=x,y,z\right)$ of the gel against the temperature for $h_{L}=0.01$.

**Figure 8 gels-06-00040-f008:**
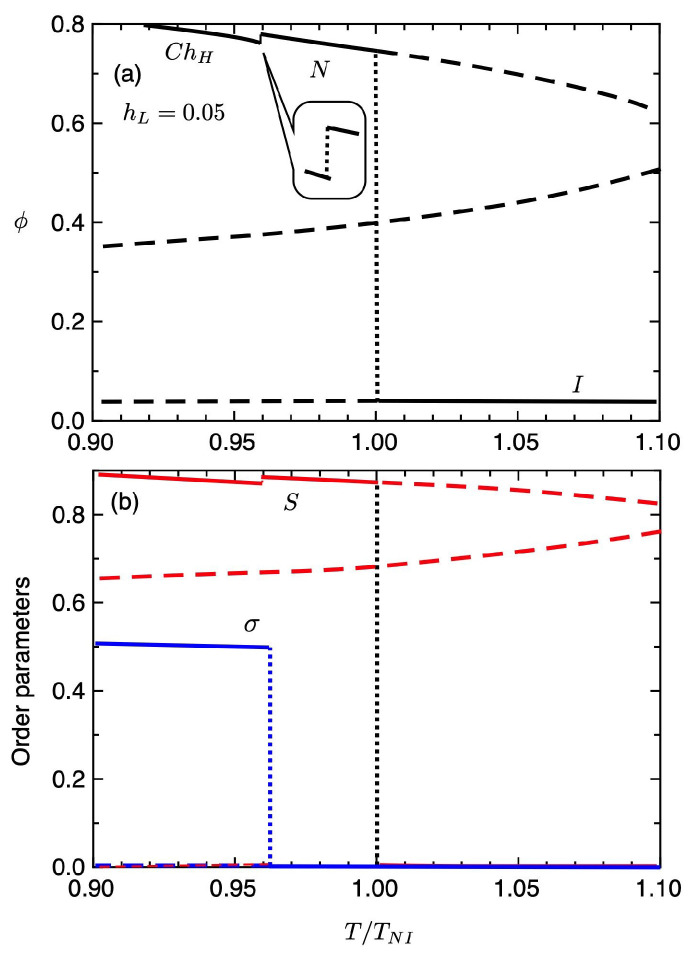
Equilibrium volume fraction of the gel (**a**) and the order parameters (**b**) plotted against the temperature $T/T_{NI}$ under the strong external field $h_{L}=0.05$.

**Figure 9 gels-06-00040-f009:**
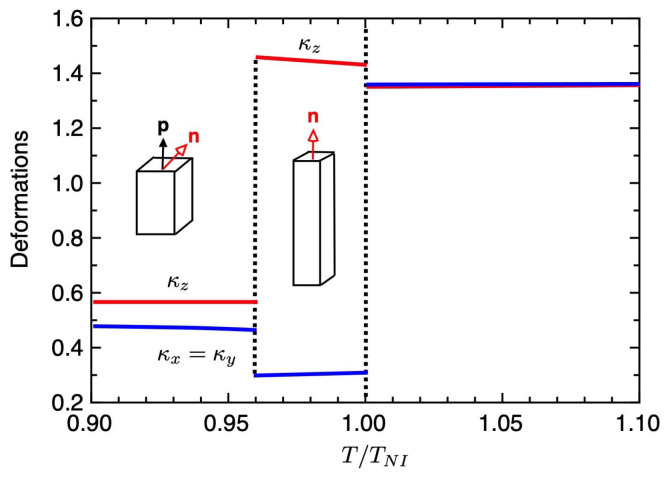
Deformations $\kappa_{i}\left(i=x,y,z\right)$ of the gel against the temperature for $h_{L}=0.05$.

**Figure 10 gels-06-00040-f010:**
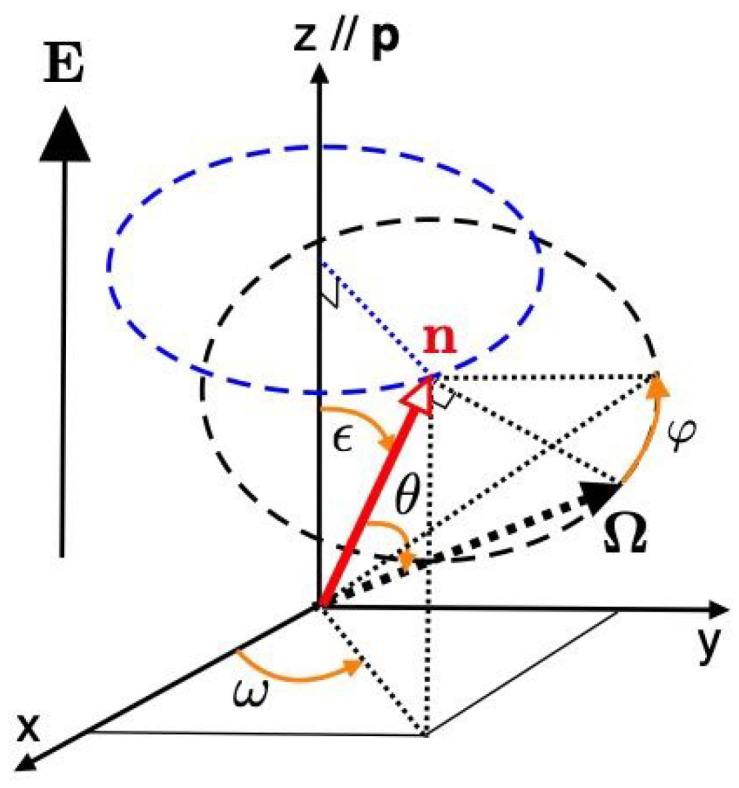
Schematic representation of the director n of a heliconical cholesteric phase at a position *z* on the same coordinate system with Figure 1. The cholesteric pitch p is parallel to the *z* axis. The orientation vector Ω of a mesogen on a subchain has the angle θ between the vector Ω and the director n. The angle φ corresponds to a rotation angle around the director n.

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
