# Peer review of "Volume Phase Transitions of Heliconical Cholesteric Gels under an External Field along the Helix Axis"

_gels, 2020, doi:10.3390/gels6040040_

Round 1

Reviewer 1 Report

The manuscript by Matsuyama reports a mean field theory for describing the behavior of cholesteric elastomers and gels which are subjected to the external field. The numerical calculation showed that the application of the external field enables various phase transitions and deformations of materials. The work is original and contains sufficient quality. I recommend the manuscript for publication after correcting minor grammar mistakes.

Author Response

I greatly appreciate the extensive comments for referee.

Reviewer 2 Report

See the attached PDF file.

Author Response

I greatly appreciate the extensive comments of  the reviewer 2. I have addressed each point, as listed below (see the attached file). The corresponding modifications to the manuscript appear in yellow for easy identification.

Reviewer 3 Report

This paper extends on earlier work by the author, now including an heliconical structure of the cholesteric phase. The results in this paper are new and interesting and deserve to be published. Before I can recommend publication, however, the authors should address the following remarks:

1) I am not sure why the system under consideration is referred to as a “gel”. The system under consideration in this paper is clearly an equilibrium system (allowing the described minimization of the free energies to obtain its structure), contrary to a gel which is an arrested, long-lived non-equilibrium state (due to strong attractive interactions between its constituents).

2) The nematic free energy of the mesogens (like in eq.35) is of the Onsager type, and accounts for mesogen interactions (in a mean-field Maier-Saupe approximation) and orientational contributions to the free energy as if the mesogens were present in bulk as independent identities, without the constraint that they are bounded to the polymer backbone. Can anything be said about the contribution to the mesogen free energy due to the constraint that they are bounded to the polymer backbone? And, are the mesogens allowed to freely rotate in directions perpendicular to the local orientation of the back bone? In Fig.1, the mesogens seem to be all oriented within the plane spanned by the backbone. Please explain.

3) Although a reference for the derivation of eq.37 is given, it would be good to have a short explanation how this derivation goes, and what the physical interpretation of eq.37 is.

4) Please add (possibly in an appendix) the derivation of eq.41-43. In the original Onsager theory for bulk nematics of long and thin rods, it has not been possible to derive an exact expression for the orientational pdf. I wonder how the authors managed to obtain an analytical expression for their much more complex system as considered by Onsager.

5) It would increase the readability of the paper if all the sizes and numbers of segments etc. introduced in the first part section 2 could be highlighted in an additional figure.

6) \beta in eq.5 should be defined (not first after eq.6).

7) The results are really interesting. The question is whether there are any experimental results known with which a comparison can be made.

8) Typos: line 47: “defamation”.

just below eq.15: “random work”.

line 67: end of the sentence “[]”.

line 144: “[]”.

In view of the above comments I recommend publication after (major) revisions.

Author Response

I greatly appreciate the extensive comments of  the reviewer 3. I have addressed each point, as listed below (see the attached file). The corresponding modifications to the manuscript appear in yellow for easy identification.

Round 2

Author Response

Thank you.

Akihiko Matsuyama

Reviewer 3 Report

I am not satisfied with the response to my first report concerning the points 2, 3, and 4. I can only recommend publication when the author addresses these points in a more satisfactorily fashion:  

Previous point 2) No explanation is given as to why a (orientational) bulk free energy can be used for the mesogens, despite the constraint that these are bounded to the polymer backbone. The author simply states that “we assume that the interaction parameter \nu_L includes the effects of these constrained mesogens”, but no arguments are given why that is reasonable. Such arguments should be added to the paper. Similarly, simply restating that the mesogens can indeed freely rotate, despite the fact that they are bounded to the polymer backbone, does not answer my question. The author should explain in the paper why the mesogens can be assumed to freely rotate despite the fact that they are bound to the polymer backbone. What is it about the chemical structure that the mesogens are able to freely rotate in 3D?

Previous point 3) Eq.37 contains the elastic constants k_2 and k_3, which do not appear in Eqs.31, 32,  and 34. Eq.37 is therefore not obtained by “Substituting the tensor order parameter Eq.31 into Eq.32”. Please explain in more detail how eq.37 has been obtained from the previous equations in the paper, and how the elastic constants come into play.

Previous point 4) In Eqs.A2-A4, the functions f and P_2 do not appear in Eqs.A2-A4 as such, but only in integrated form. I do therefore still not understand how Eq.41 is obtained (equality of integrals does not imply equality of the integrands). But maybe I am missing something. Please explicitly perform the last step of the supposed derivation of Eq.41 in Appendix A.

Round 3

Reviewer 2 Report

  This manuscript is publishable after the following issues are addressed. However, I strongly recommend the author to check the manuscript thoroughly to eliminate trivial mistakes such as “stabe” in line 169 (I was really irritated to find such mistakes in every round of review).

* The left hand side of eq. (53), the total derivative, should be $d f_e/d\phi$, not $\partial f_e/\partial\phi$.

* in eq. (54), are the derivatives with respect to $f(\tilde\zeta)$ functional derivatives that should be represented as $\delta f_e/\delta f(\tilde\zeta)$?

* In line 163, I could not understand what the author means by “Eq (54) satisfies Eq. (45).”

* Do Figs. 7 and 8 really concern $T/T_{CI}$, not $T/T_{NI}$?

Reviewer 3 Report

I thank the authors for the chances they made. The paper is now ready for publication.

Author Response

Thank you for your reviewing.